# Circulation-regulated impacts of aerosol pollution on urban heat island in Beijing

Fan Wang[1], Gregory R. Carmichael[2], Jing Wang[3], Bin Chen[4], Bo Huang[5], Yuguo Li[6], Yuanjian Yang[7], Meng Gao[1,8]

[1]Department of Geography, State Key Laboratory of Environmental and Biological Analysis, Hong Kong Baptist University, Hong Kong SAR, 999077, China

[2]Department of Chemical and Biochemical Engineering, The University of Iowa, Iowa City, IA 52242, USA

[3]Tianjin Key Laboratory for Oceanic Meteorology, and Tianjin Institute of Meteorological Science, Tianjin 300074, China

[4]Division of Landscape Architecture, Faculty of Architecture, The University of Hong Kong, Hong Kong SAR, 999077, China

[5]Institute of Space and Earth Information Science and Department of Geography and Resource Management, The Chinese University of Hong Kong, Hong Kong SAR, 999077, China

[6]Department of Mechanical Engineering, The University of Hong Kong, Pokfulam, Hong Kong SAR, 999077, China

[7]Collaborative Innovation Centre on Forecast and Evaluation of Meteorological Disasters, Key Laboratory for Aerosol-Cloud-Precipitation of China Meteorological Administration, School of Atmospheric Physics, Nanjing University of Information Science and Technology, Nanjing 210044, China

[8] Southern Marine Science and Engineering Guangdong Laboratory (Guangzhou), Guangzhou 511458, China

*Correspondence to*: Meng Gao (mmgao2@hkbu.edu.hk)

**Abstract.** Unprecedented urbanization in China has led to serious urban heat island (UHI) issues, exerting intense heat stress on urban residents. Based on observed temperature and $PM_{2.5}$ concentrations in Beijing over 2016-2020, we find diverse influences of aerosol pollution on urban heat island intensity (UHII) under different circulations. When northerly winds are prevalent in urban Beijing, UHII tends to be much higher in both daytime and nighttime and it is less affected by aerosol concentration. However, when southerly and westerly winds are dominant in rural Beijing, UHII is significantly reduced by aerosol pollution. Using coupled aerosol-radiation-weather simulations, we demonstrate the underlying physical mechanism, which is associated with local circulation and resulting spatial distribution of aerosols. Our results also highlight the role of black carbon in aggravating UHI, especially during nighttime. It could thus be targeted for cooperative management of heat islands and aerosol pollution.

## 1 Introduction

The dramatic global rise of urbanization has led a rapid growth of urban population (Elmqvist et al., 2013) and a quick enlargement of urban sizes (Seto et al., 2012). Massive use of cement and asphalt in urban construction changes local topography and thermal properties of urban surfaces (Mohajerani et al., 2017; Voogt and Oke, 2010). Coupled with elevated

anthropogenic heat and air pollutants from booming human activities, expansion of impervious surface exacerbates urban warming (Grimmond, 2007; Oke, 1982) and degrades diffusion of pollutants (Lewis, 2018; Olivier et al., 2020; Seinfeld, 1989; Zhao et al., 2021), leading to a series of environmental and social issues (Kumar et al., 2017; Mcdonough et al., 2020). Urbanization has been demonstrated as a critical factor contributing to global warming (Argüeso et al., 2013; Sun et al., 2016; Wilke et al., 2019) and more frequent occurrences of extreme high-temperature events (Sun et al., 2019; Wang and Wang, 2021; Xiao et al., 2022; Zhou et al., 2020). Emissions of trace gases and particles from transportation, industries, and residential activities also threaten health and wellbeing of urban residents (Crutzen, 2004; Salma et al., 2015; Wilke et al., 2019).

Different surface properties generated by urbanization makes cities warmer than surrounding areas, and urban heat island (UHI) is thus created by such thermal gradients (Oke, 1973). UHI is usually calculated as the temperature difference between urban and surrounding rural areas (Deilami et al., 2018). It increases frequency and intensity of heat waves in urban areas, and thus aggravates heat stress on urban residents (Cao et al., 2016; Li and Bou-Zeid, 2013; Santamouris, 2014). The intensity of urban heat island (UHII) is influenced by multiple factors, including ground energy balance, anthropogenic heat release and sky view factor (Oke and Stewart, 2012; Xie et al., 2016a; Xie et al., 2016b). Air pollutants especially aerosols modify surface radiation balance through aerosol radiative effects (ARE), exerting potential impacts on UHII (Cao et al., 2016). ARE cuts amount of downward shortwave radiation (SWD) reaching the ground, reduces sensible heat (SH) flux, and lowers height of the planet boundary layer (PBLH) (Satheesh and Krishnamoorthy, 2005; Yu et al., 2006), which aggravates severity of haze events in China (aerosol-radiation feedback, hereafter as ARF, Ding et al., 2016; Gao et al., 2016b; Wu et al., 2019b; Zhao et al., 2017). The impacts of aerosols on UHI vary with locations, seasons, and day/night (Han et al., 2020). Urban is usually the center of pollution with relatively higher aerosol concentrations than rural areas (Seinfeld, 1989). Under this circumstance, aerosol can cut down more downward shortwave radiation and result in stronger reduction on near surface temperature in urban areas, which reduces UHII in daytime (Li et al., 2018; Li et al., 2020a; Longxun et al., 2003; Sang et al., 2000; Yang et al., 2020). However, absorbing aerosols (e.g., black carbon, BC) absorb and release radiation to increase longwave radiation energy received on urban surface, resulting in intensified UHI, especially during nighttime (Cao et al., 2016; Chen et al., 2018; Zheng et al., 2018).

China has been experiencing unprecedented urbanization over the past four decades (Gong et al., 2020; Guan et al., 2018). As the capital city, Beijing has achieved a high level of urbanization (Wang et al., 2019; Zhou et al., 2021), leading to serious UHI (Miao et al., 2009; Yang et al., 2013). Although the association between aerosol pollution and UHII in Beijing has been realized, no consensus has been reached(Cao et al., 2016; Li et al., 2020b; Yang et al., 2021; Yang et al., 2020; Yu et al., 2020; Zheng et al., 2018). Yang et al. (2020) and Zheng et al. (2018) found weakened UHI in winter by aerosols in daytime but enhanced during nighttime. Li et al. (2020b) argued that aerosol concentrations in southern rural areas are usually higher than those in urban or northern rural areas of Beijing, causing a southward shift of UHI. However, Yang et al. (2021) claimed that aerosols increased urban heat island intensity in winter in Beijing in daytime but weakened it during nighttime. The contradictory results are partly due to selection of urban and rural monitoring stations, and a detailed

explanation with numerical experiments is still lacking. Beijing is located in the North China Plain (NCP), with the Yan Mountains to the northwest and the Bohai Gulf to the southeast. The special topography induced local circulation patterns, such as foehn wind and sea breeze, complicate spatial distribution of near-surface air temperature and aerosol pollution, and thus the influences of aerosol pollution on UHI (Bei et al., 2018; Li et al., 2020c; Wang et al., 2020a). In this study, we aim to better understand how aerosol pollution affects UHI in Beijing using observations over 2016-2020 and a coupled meteorology-chemistry model. The results would offer valuable information on cooperative management of heat islands and pollution in China.

## 2 Data and Methods

### 2.1 Observational data

Observed daily average, maximum and minimum temperature, and wind speed/direction from automatic weather stations (AWS) in Beijing over 2016–2020 were obtained from the National Meteorological Information Center (NMIC), China Meteorological Administration (CMA). Preliminary quality control was implemented by the NMIC, and potential wrong records had been checked and corrected (Ren and Xiong, 2007; Ren et al., 2015). We chose two urban stations, Haidian and Guanxiangtai, and five rural stations including Huairou, Shangdianzi, Pinggu, Yanqing and Xiayunling (Fig. S1 and Table S1) to characterize UHII in Beijing. Hourly $PM_{2.5}$ concentrations over the same period were taken from the China National Environmental Monitoring Center (CNEMC) network.

### 2.2 WRF-Chem model configuration

In this study, we used the Weather Research and Forecasting model coupled with Chemistry (WRF-Chem) version 3.6.1 to explore formation of aerosols and their interactions with radiation and weather (Grell et al., 2005). We configured three domains with grid resolutions of 81 km, 27 km and 9 km. To better capture the actual land use types, we used the Moderate Resolution Imaging Spectroradiometer (MODIS) land cover data in 2010 and 2018 (Fig. S1). We used Carbon-Bond Mechanism version Z (CBMZ, Zaveri and Peters, 1999) and the 8-bin version of Model for Simulating Aerosol Interactions and Chemistry (MOSAIC, Zaveri et al., 2008) to simulate gas phase and aerosol chemistry. We added heterogeneous reactions (Gao et al., 2016a) to solve the problem of underestimation of sulfate. For other options, we followed Wang et al. (2022) to use Lin cloud microphysics (Lin et al., 1983) for cloud microphysics, Grell 3D Ensemble Scheme (Grell, 1993) for precipitation, RRTM (Mlawer et al., 1997) for sub-grid long-wave radiation and Goddard (Chou et al., 1998) for and short-wave radiation. We also used Noah land surface model (Tewari et al., 2004) for land-atmosphere exchange, Yonsei University planetary boundary layer parameterization (Noh et al., 2006) for boundary layer processes, and Urban Canopy Model (UCM, Chen et al., 2011) to include three-dimensional city structure and associated energy balance. For anthropogenic emissions, we used monthly 0.25°×0.25° Multi-resolution Emission Inventory for China (MEIC 2010 and MEIC 2018) (Li et al., 2017) in 2010 and 2018. Biogenic emissions were estimated online using the Model of Emissions of

Gases and Aerosols from Nature (MEGAN) (Guenther et al., 2006), and we did not include open biomass burning as it was not significant in Beijing during our study period (Gao et al., 2016b). Meteorological initial and boundary conditions were taken from the 6-hourly National Centers of Environmental Prediction (NCEP) Final Analysis (FNL) dataset.

    The difference in heat storage is one important factor that affects the diurnal variation of UHII. In WRF-Chem model, heat storage is calculated with land surface model, and we applied Noah land surface scheme for non-urban grids and Urban
Canopy model for urban grids. In Noah land surface scheme, heat storage is calculated using the following equations:

$$G = (1 - F_{veg})G_b + F_{veg}G_v \tag{1}$$

$$G_b = \frac{2\lambda_{isno+1}}{\Delta z_{isno+1}}(T_{g,b} - T_{isno+1}) \tag{2}$$

$$G_v = \frac{2\lambda_{isno+1}}{\Delta z_{isno+1}}(T_{g,v} - T_{isno+1}) \tag{3}$$

where $F_{veg}$ denotes fractional vegetated area, $G_b$ and $G_v$ are heat storage for bare ground and vegetated ground, respectively,
and $\lambda_{isno+1}$ represents thermal conductivity of the surface layer (snow or soil); $z_{isno+1}$ is layer thickness of the surface layer (snow or soil), $T_{isno+1}$ represents temperature of the surface layer of snow (under $isno +1 < 0$) or soil (under $isno = 0$), and $T_{g,b}$ and $T_{g,v}$ stand for ground surface temperature at bare ground fraction and vegetated fraction, respectively. In Urban Canopy model, heat storage is calculated using

$$G = G_0 + 2\int_0^{z_r} \left[\frac{\partial(\rho_b c_b T_b)}{\partial t}\right] d_z \tag{4}$$

where $G_0$ denotes the surface heat flux into the ground per unit area, and $\rho_b$, $c_b$, and $T_b$ represent density, specific heat, and temperature of buildings.

**2.3 Experimental design**

    We designed two groups of simulations of a severe haze event in the winter of 2010 (Case_2010) and a light pollution event in the spring of 2018 (Case_2018). Each of them had four sets of simulations, namely AF, NAF, NBC and Ndust, to
explore the impacts of aerosols on UHII, including roles of scattering and absorbing aerosols. AF cases were performed with actual conditions, while we turned off aerosol-radiation feedbacks in NAF. NBC was designed as the simulation that ignored the absorption of black carbon (BC) and absorption of dust was turned off in Ndust. In Case_2010, the simulation period covered from 11 to 20 January 2010 with first five days as spin-up. The study period included three days and nights from 8:00 LST on 16 January to 8:00 LST on 19 January 2010. It covered an entire severe haze pollution event in winter, during
which UHI was formed and wind direction changed over days, offering conditions to analyze the impacts of ARF on UHII under different circulation conditions. In Case_2018, the simulation period covered from 19 to 28 April 2018 with first five days as model spin-up time. The study period was from 7:00 LST on 24 April to 7:00 LST on 27 April 2018. It covered a light aerosol pollution in spring and was used to evaluate if the impacts of aerosols on UHII are consistent under different seasons and aerosol pollution conditions. As changes from turning off absorption of dust were negligible, we did not show
results from Ndust in figures.

**2.4 Calculation of UHII**

We defined $UHII_{obs}$ as observed differences in average 2m air temperature ($T_{2m}$) between all urban stations and all rural stations. Following Yang et al. (2020), we also calculated $UHII_{max}$ and $UHII_{min}$ as differences in daily maximum temperature ($T_{max}$) and daily minimum temperature ($T_{min}$). As $T_{max}$ often appears in the afternoon and $T_{min}$ usually happens at late night or early morning before sunrise, we used $UHII_{max}$ and $UHII_{min}$ to refer to daytime and nighttime UHII. For simulated UHII, we defined $UHII_{sim}$ as the difference in average $T_{2m}$ between urban areas and a buffer zone around the urban area that has the same size as the urban area, which is similar to that adopted with satellite products in Zhou et al. (2014). We chose these two different definitions of UHII for observation and simulation to evaluate uncertainty induced by the spatial limitation of monitoring stations.

**3 Results and discussions**

**3.1 Observational evidence of circulation-regulated impacts of aerosol pollution on UHII**

Fig. 1 presents the probability distributions of UHII under different $PM_{2.5}$ concentrations. On clean days (daily average $PM_{2.5}$ concentration below 75 µg m$^{-3}$), the distribution of UHII tends to be more towards larger values with a mean of 2.34 K. It decreases to 1.8 K on pollution days (daily average $PM_{2.5}$ concentration above 75 µg m$^{-3}$) (Fig. 1a, d). UHII exhibits higher values at nighttime than those in daytime. In both daytime and nighttime, $PM_{2.5}$ pollution is associated with decreased UHII in Beijing. In this analysis, we calculated mean $PM_{2.5}$ concentration of all stations (Table S2) within Beijing and used it to determine if Beijing is polluted (daily mean $PM_{2.5}$ concentration $\geq$ 75 µg m$^{-3}$) or clean (daily mean $PM_{2.5}$ concentration < 75 µg m$^{-3}$). We also examined the distribution of daily mean urban and rural $PM_{2.5}$ concentrations under clean and polluted conditions (Table S3), and we found that 17.07 % of clean days of rural stations were classified as polluted ones of urban stations due to the pollution gradient between urban and rural areas. However, these misclassified days were mostly light polluted with $PM_{2.5}$ concentrations over 60 µg m$^{-3}$ (Table S3).

We also evaluated how different standards of polluted or clean would affect results, and we included results based on the standard that $PM_{2.5}$ concentrations of all stations in Beijing meet the thresholds of clean or polluted (Fig. S2) and results based on the standard that both average $PM_{2.5}$ concentrations of all urban stations and rural stations meet the criterion (Fig. S3). Compared with Fig. 1 using mean $PM_{2.5}$ concentration of all stations, we found similar distributions and negligible differences in mean values. When $PM_{2.5}$ concentrations of all stations meet the criterion, we found the mean values increased by 0.03-0.04 K for clean conditions but decreased by 0.14 K during daytime and 0.06 K during nighttime. When we used average $PM_{2.5}$ concentrations of all urban stations and rural stations to determine clean or polluted, mean values decreased by 0.01 K for clean conditions and increased by 0.01 K and 0.06 K during daytime and nighttime, respectively. We thus believe that using the daily mean $PM_{2.5}$ concentrations averaged over all stations can properly represent the regional feature of aerosol pollution and would not affect our findings.

It was found previously that aerosol pollution led to decreased $UHII_{max}$ (daytime) but increased $UHII_{min}$ (nighttime) (Yang et al., 2021; Yang et al., 2020). This discrepancy is associated with the differences in regions that considered as rural in the calculation. We used rural stations located in the west and north of Beijing as rural in the calculation of UHII, and $PM_{2.5}$ concentrations are usually much lower there. As a result, temperature at these rural stations is less affected by aerosol pollution. We designed a simplified flow chart to show how UHII is changed in daytime and nighttime, assuming that rural areas are not influenced by ARE (Fig. 2). ARE reduces near surface temperature in urban areas, leading to a weakened UHII and heat storage throughout the day. Although the strengthened longwave radiation process in nighttime that due to absorption of aerosols in daytime alleviates the reduction of temperature in urban areas, decreased daytime temperature and heat storage release contributes more to near surface temperature and results in weakened UHII. The increase of UHII due to strengthened longwave radiation process is smaller than the decrease of UHII caused by reduced temperature and heat storage release during daytime (see difference between Fig. 1b, c, e and f).

Fig. 3 displays UHII under different wind directions and $PM_{2.5}$ pollution. As wind direction usually differs in urban areas and rural areas in the west and north of Beijing (Chen et al., 2017), we discuss separately based on the wind direction in urban sites and rural sites. We observe elevated UHII when northerly winds are prevalent in urban areas on polluted days (Fig. 3a, c). The mean UHIIs are 2.0 and 1.8 K in daytime and 2.9 and 2.8 K in nighttime on clean and polluted days, respectively. This is associated with reduced aerosol concentrations in urban regions by northerly winds in urban areas (Table 1). From clean to polluted conditions under northerly, lower reduction in UHII by aerosols is accordingly found (Fig. 3). Larger decreases in UHII in daytime can be found from clean to polluted conditions under easterly, southerly and westerly winds conditions, and these decreases are weakened at nighttime. The weakening may be caused by the longwave radiation process as absorptive aerosols release heat during night to alleviate decreases in surface temperature, especially in urban areas (Cao et al., 2016; Yang et al., 2020). This process has also been confirmed with our simulation that ARE-induced enhanced longwave radiation reduces the weakening of UHII in nighttime (Fig. S4).

When sorted by wind directions in rural areas, we still find strongest UHII under northerly wind conditions (Fig. 3b, d). However, UHII is relatively weak and the probability of "cold islands" in daytime increases when westerly or southerly winds are prevalent. The weak UHII under westerly wind condition is associated with foehn wind that northwesterly or westerly travel through the Yan Mountains, as foehn wind is able to heat rural areas and reduce the urban-rural thermal gradients (Ma et al., 2013). When southerly winds are prevalent, warmer southerly winds from lower latitude tend to heat southern rural areas faster than urban due to blocking of air by buildings and larger heat capacities of urban impervious surface and buildings. We also detect larger reductions of UHII by aerosols when westerly or southerly winds are dominant (Fig. 3b, d), suggesting that foehn wind and warm southerly winds are likely to amplify the weakening effect of aerosols on UHII.

**3.2 Diurnal variations of the impacts of ARE on UHII**

Although we identified above consistent weakening of UHII by aerosols during both daytime and nighttime, the influences vary with wind directions, which are regulated by background circulation patterns. To understand the underlying mechanism of the varying influences and to reduce uncertainty induced by selection of monitoring stations, we conducted model simulations of a typical haze event that occurred in winter in Beijing (Gao et al., 2016b) since aerosol concentrations are usually higher in winter in Beijing (Gao et al., 2018). We also designed simulations of a light pollution event in spring to evaluate if the results are robust under different seasons and aerosol pollution conditions. As the aim of this section is to explore the underlying mechanism of interactions between aerosol pollution and urban heat island, although the period differs from observations shown above, the selected cases are sufficient to represent the observed varying wind conditions. Model configurations in this study follow Gao et al. (2016b), and extensive model evaluations using multi-source observations indicated reliable reproduction of the wintertime haze event (Case_2010) by WRF-Chem. We additionally evaluated the performance of WRF-Chem in simulating Case_2018 (Fig. S6 and Table S4), and similar results were yielded. Further validation of the ability of the model to simulate site-based UHII is shown in Fig. S7 and Fig. S8. The model successfully reproduces the temporal variation of UHII in Beijing, and differences in values are generally within the trusted range, compared with previous simulations (Li and Bou-Zeid, 2013; Miao et al., 2009). To better clarify the influence induced by selection of rural areas, we added Fig. S9 to show the simulated UHII calculated based on site locations and area average. Apparent difference can be found that site-based ΔUHII (difference due to ARE) decreases more than area-based UHII especially in nighttime because of lower $PM_{2.5}$ concentrations in the rural sites than selected rural area.

Fig. 4 shows the temporal variation of UHII of three cases, namely AF, NAF and NBC in Case_2010. Given the negligible contribution of absorption of dust to UHII, the results from the Ndust case are not shown. The impacts of ARE on UHII exhibit a bimodal distribution during daytime (Fig. 4a). The first peak and valley appear after sunrise, and the second peak and valley occur before sunset. These variations are associated with the fact that changes in $T_{2m}$ occur earlier in rural areas. Aerosol pollution cuts down SWD in both urban and rural areas (Fig. S10a, b) after sunrise. Near surface temperature in rural areas usually increases faster than that of urban areas (Oke, 1982). As a result, temperature in rural areas exhibits earlier declines in response to ARE, as indicated by ARE induced changes in $T_{2m}$ in Fig. 4b, d, f, g. The second peak is caused by the similar reason that ARE results in the earlier decrease in $T_{2m}$ of rural areas (Fig. 4b, f), but the second peak is also contributed by release of heat storage. Heat storage of rural areas is usually lower than that of urban areas, yet heat is released more slowly in rural areas, as suggested in Fig. S10 that heat storage is smaller in daytime but reaches zero later than it in urban areas (Fig. S10c, d). Heat storage release contributes to upward sensible heat flux at the ground, which further increases $T_{2m}$ after midday and slows down the decreases after the peak in the afternoon (Oke et al., 1992). As a result, a faster declining of $T_{2m}$ in rural areas is found than that in urban areas (Fig. 4b, f). ARE reduces heat storage in both rural and urban areas, and the smaller heat storage and slower release of heat in rural areas make $T_{2m}$ decrease earlier, leading to the second peak and valley. Fig. 5 shows related results for the Case_2018 case, and we find that ARE generally reduces

UHII except on 26 April. The bimodal distribution during daytime still exists but is inconspicuous. This is because the lower ambient $PM_{2.5}$ concentrations in the spring of 2018 reduce the gradient between urban and rural areas, and weakens the impacts of ARE on shortwave radiation and near-ground air temperature.

### 3.3 Diverse influences of ARE on UHII and the role of local circulation

We label days and nights of the study period as D1, N1, D2, N2, D3, N3 in Case_2010 and D4, N4, D5, N5, D6, N6 in Case_2018 in order, and find diverse influences under different wind patterns. On D1 and N1, we observe that ARE weakens UHII by 0 - 0.4 K if the absorption of BC is not considered, due to larger amount of scattering aerosols in urban areas (Cao et al., 2016; Yang et al., 2020). The weakening is larger at daytime and UHII is enhanced at nighttime when absorption of BC is considered (Fig. 4a). BC is potent in absorbing radiation, and it causes larger decrease in SWD at daytime. BC also
warms the atmosphere which increases downward longwave radiation (Fig. S11 and Fig. S12) in nighttime (Cao et al., 2016; Zheng et al., 2018). On D2, a cold island with an intensity of ~-0.8 K is formed in Beijing, and ARE enhances the intensity of cold island. Due to the large reduction of UHII by aerosols at daytime (Fig. 4a), we still find negative effect of ARE on UHII on N2. Yet the negative effects weaken and become positive before sunrise. Different from previous two days, ARE enhances UHII with a maximum value of 1 K on D3. This is associated with reduced differences in $PM_{2.5}$ concentrations
between urban and rural areas on D3 (Fig. 4c and Fig. S13c). The conditions on N3 are similar with those on N2. The impacts of aerosols on UHII in nighttime are mainly generated by modified downward longwave radiation (Yang et al., 2021; Zheng et al., 2018), which influences the UHII maintained after sunset. BC is the main light-absorbing aerosol (Gao et al., 2021; Ramanathan and Carmichael, 2008), and higher concentrations of BC (Fig. S11) lead to enhanced UHII in nighttime (Fig. S12). This explains the larger intensified UHII (~2 K) on N2. On D4 and D5, due to much lower $PM_{2.5}$
concentration, ARE reduces UHII by less than 0.2 K (Fig. 5). The lower concentration also diminishes absorption of shortwave radiation during daytime, which further reduces downward longwave radiation and causes weakened UHII on N4 and N5. On N5, we find a sudden ARE-induced increase in UHII (Fig. 5), and it is associated with the elevated $PM_{2.5}$ concentration on N5 (Fig. 5c).

    The above-mentioned diverse influences on different days of the study period are mainly controlled by local circulation.
Fig. 6 presents spatial distributions of daytime 2m air temperature and 10m wind fields over the study period. On D1 (Fig. 6a, d, g), southerly winds dominate the NCP, bringing warmer air to Beijing. However, due to relatively higher $PM_{2.5}$ concentrations in the south of Beijing (Fig. S13), ARE decreases $T_{2m}$ as well as wind speeds. As a result, the warmer air transported from the southern regions to the south of Beijing is weakened, and only southern rural areas can be significantly heated, reducing the UHII of Beijing. This explains why UHII tends to be relatively weaker and larger reductions of UHII by
aerosols when southerly winds are prevalent in NCP (Fig. 3). On D2 (Fig. 6b, e, h), strong northwesterly winds (foehn wind) influence Beijing, and the entire western suburbs of Beijing heat up rapidly, forming a cold island. Meanwhile, mountains block strong northwesterly winds, and wind speeds on NCP are relatively weak, favoring accumulation of aerosols in urban areas (Fig. S13). Accordingly, ARE significantly reduces $T_{2m}$ in urban areas and further inhibits the UHII in the west of the

city, consistent with the results shown in Fig. 3b that largest reductions in UHII was caused by aerosol pollution. On D3 (Fig. 6c, f, i), we detect a southeasterly sea breeze coming from the Bohai Gulf. Under the influence of the Yan Mountains, wind directions change to northeasterly when they reach Beijing. Consequently, more aerosols accumulate in the southern rural areas of Beijing (Fig. S13), ARE contributes to larger decrease in $T_{2m}$ in rural areas than that in urban areas. We thus observe an enhanced UHII caused by ARE on that day (Fig. 4a). This situation is consistent with observations that strongest UHII and alleviated reduction of UHII by aerosol pollution occur when urban areas are under northerly winds (Fig. 3a, c). When light pollution event happens, similar responses (except results on N5) but smaller values are found (Fig. S14). The identified sudden ARE-induced increase in UHII on N5 (see Fig. 5) is caused by southerly winds. Southerly wind transports warm air masses with high $PM_{2.5}$ concentrations from lower latitude to the north, and this process enhances the downward longwave radiation to heat the surface of urban and southern rural regions, resulting in enhanced UHII (Fig. S15). This also explains why UHII tends to decrease less when southerly winds are prevalent in nighttime (Fig. 3c).

## 4 Summary

Observed temperature and $PM_{2.5}$ concentrations in Beijing over 2016-2020 suggest that aerosol pollution is associated with decreased UHII in Beijing at both daytime and nighttime, yet the influences of aerosol pollution on UHII are diverse under different circulation patterns. When northerly winds are prevalent in urban Beijing, UHII tends to be much higher at both daytime and nighttime and it is less affected by aerosol concentration. The mean values are 2.0 (1.8) and 2.9 (2.8) K in clean (polluted) conditions in daytime and nighttime, respectively. However, when southerly and westerly winds are dominant in rural Beijing, UHII is significantly reduced by aerosol pollution by over 0.5 K. Using coupled aerosol-radiation-weather simulations, we demonstrate the underlying physical mechanism, which is associated with local circulation and resulting spatial distribution of aerosols.

Previous studies documented opposite effects of aerosol pollution on UHII in Beijing (Cao et al., 2016; Yang et al., 2021; Yang et al., 2020; Yu et al., 2020; Zheng et al., 2018), and other cities(Li et al., 2018; Li et al., 2020a; Wu et al., 2017; Wu et al., 2019a). Our study highlights that the influences of aerosol pollution on UHII vary with local circulation, which is particularly important for Beijing due to the complex topography. Besides, heat can be modulated by local circulation to influence the impacts of aerosol pollution on UHII. Therefore, investigating the dominant synoptic pattens in certain areas may contribute to a better understanding of the aerosol-UHII interactions and provide guidance for mitigation strategies (Yang et al., 2020; Yu et al., 2020). Aerosol pollution in China has been alleviated significantly since the implementation of strict clean air policies after 2013 (Gao et al., 2020; Wang et al., 2020b). Yet there is still no evidence showing that it has co-benefits of reducing UHI (Li et al., 2007; Cao et al., 2016). It was found that decreasing aerosols led to intensification of urban warming and UHI, which further contributed to aggravation of ozone pollution (Wang et al., 2020b; Yu et al., 2020). Thus, controlling aerosol pollution might even pose greater challenges for urban climate and environment management. In this study, our model experiments emphasize the role of BC in aggravating UHI, especially during nighttime (Fig. 4). It

could thus be targeted for cooperative management of heat islands and pollution. Some climate and environment friendly measures including urban greening (Chen et al., 2019; Knight et al., 2016) could be adopted further to alleviate both urban heat and air pollution, considering the evapotranspiration effects and extra green space for deposition.

## Data availability

The data used in this study can be accessed through contacting the corresponding author.

## Author contributions

MG designed the study, and FW performed model simulations and analyzed the data with help from GRC, JW, BC, BH, YL, YY; FW and MG wrote the paper with inputs from all other authors.

## Competing interests

The authors declare that they have no conflict of interest.

## Financial support

This study was supported by grants from Research Grants Council of the Hong Kong Special Administrative Region, China (project no. HKBU22201820 and HKBU12202021), National Natural Science Foundation of China (No. 42005084) and Natural Science Foundation of Guangdong Province (no. 2019A1515011633).

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

**Table 1: Average PM$_{2.5}$ concentration (unit: µg m$^{-3}$) in urban and rural areas under different prevalent wind directions.**

| Wind directions | Easterly | Southerly | Westerly | Northerly |
|---|---|---|---|---|
| Urban PM$_{2.5}$ | 58.23 | 53.88 | 52.24 | 49.49 |
| Rural PM$_{2.5}$ | 50.82 | 47.34 | 44.68 | 43.31 |

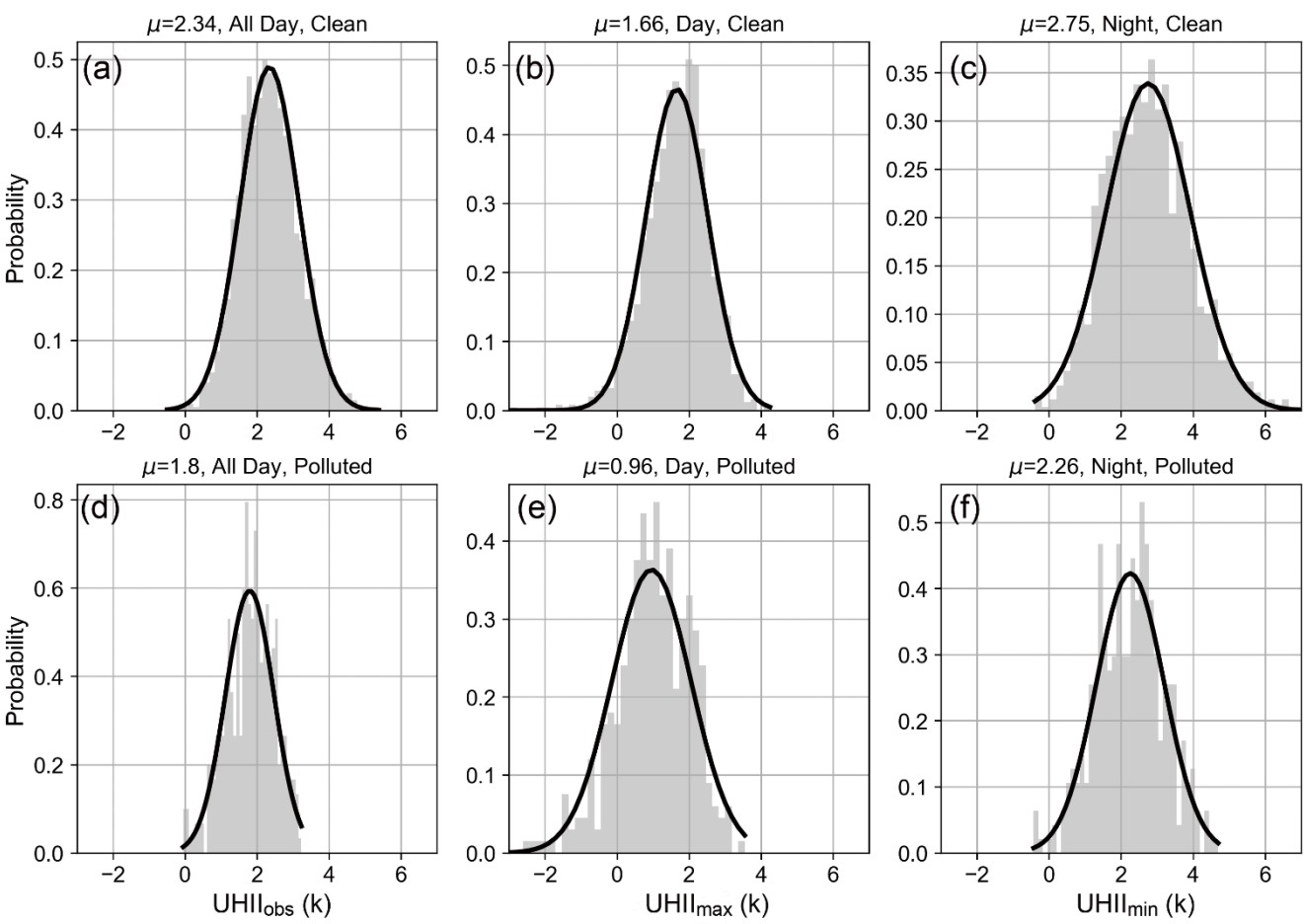

 **Figure 1: Probability distribution of UHII$_{obs}$ (a, d), UHII$_{max}$ (b, e) and UHII$_{min}$ (c, f) under different pollution conditions. The bold curve in each subgraph is normal distribution curve, and µ denotes the average value.**

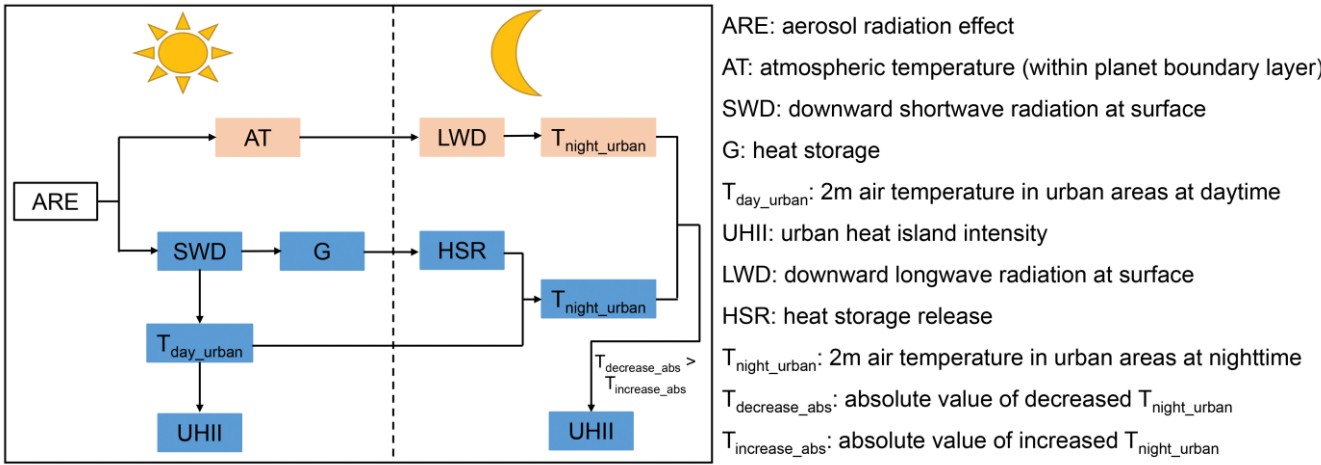

**Figure 2: Flow chart shows how UHII is changed at daytime and nighttime, assuming that rural areas are not influenced by ARE. Pink boxes show increasing trend while blue ones show decreasing trend.**

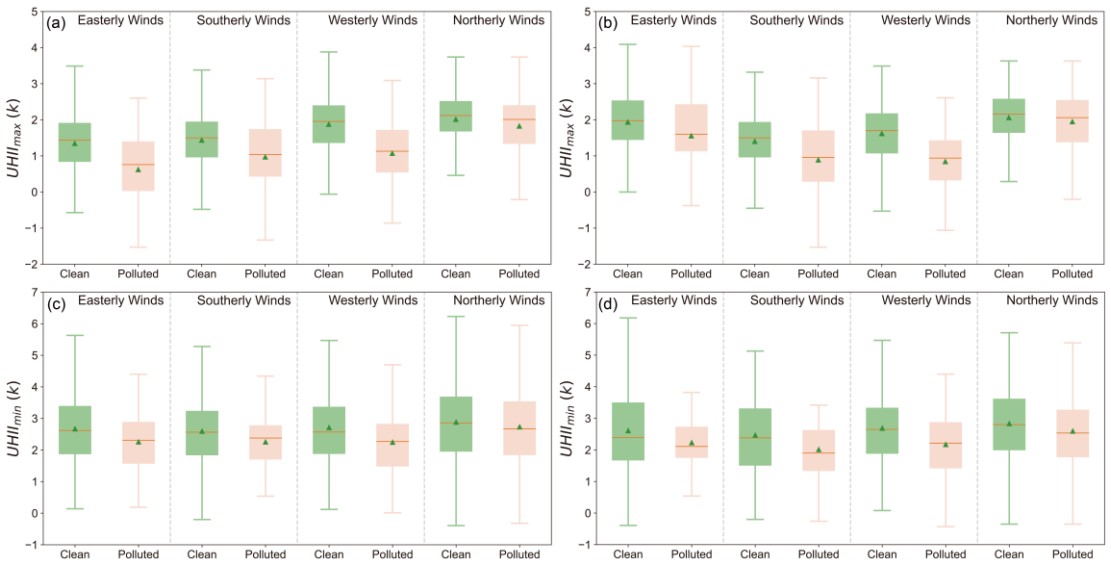

**Figure 3: Distribution of UHII under different wind and pollution conditions. Panel (a) and (c) are classified based on the wind direction in urban areas, while panel (b) and (d) are based on wind direction in rural areas. Green triangles represent average values, red lines are median values; box chart values denote mean value minus standard deviation, 25% quantile, 75% quantile, and the mean value plus standard deviation, from the bottom to up, respectively.**

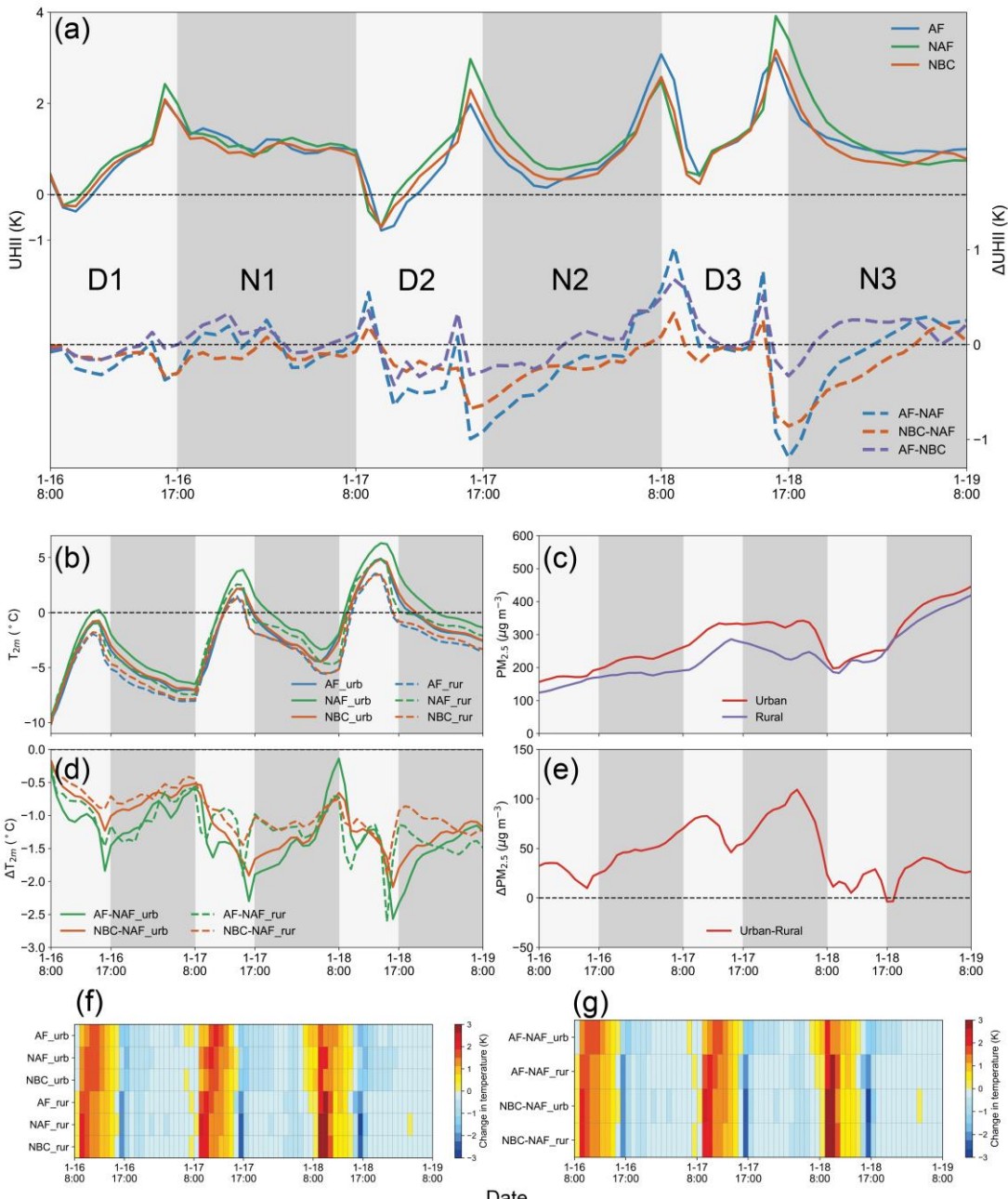

**Figure 4: Variations of UHII_sim of all cases and difference across them (a) in Case_2010. Variations of T_2m (b) and △T_2m (d) in urban and rural areas. Variations of PM2.5 (c) and △PM2.5 (e) in urban and rural areas of AF case. Hourly changes in T_2m (f) and △T_2m (g) in urban and rural areas. AF-NAF represents the influence of ARE on UHII. NBC-NAF represents the influence of ARE on UHII by all aerosols but BC. AF-NBC represents the influence of BC absorption on UHII.**

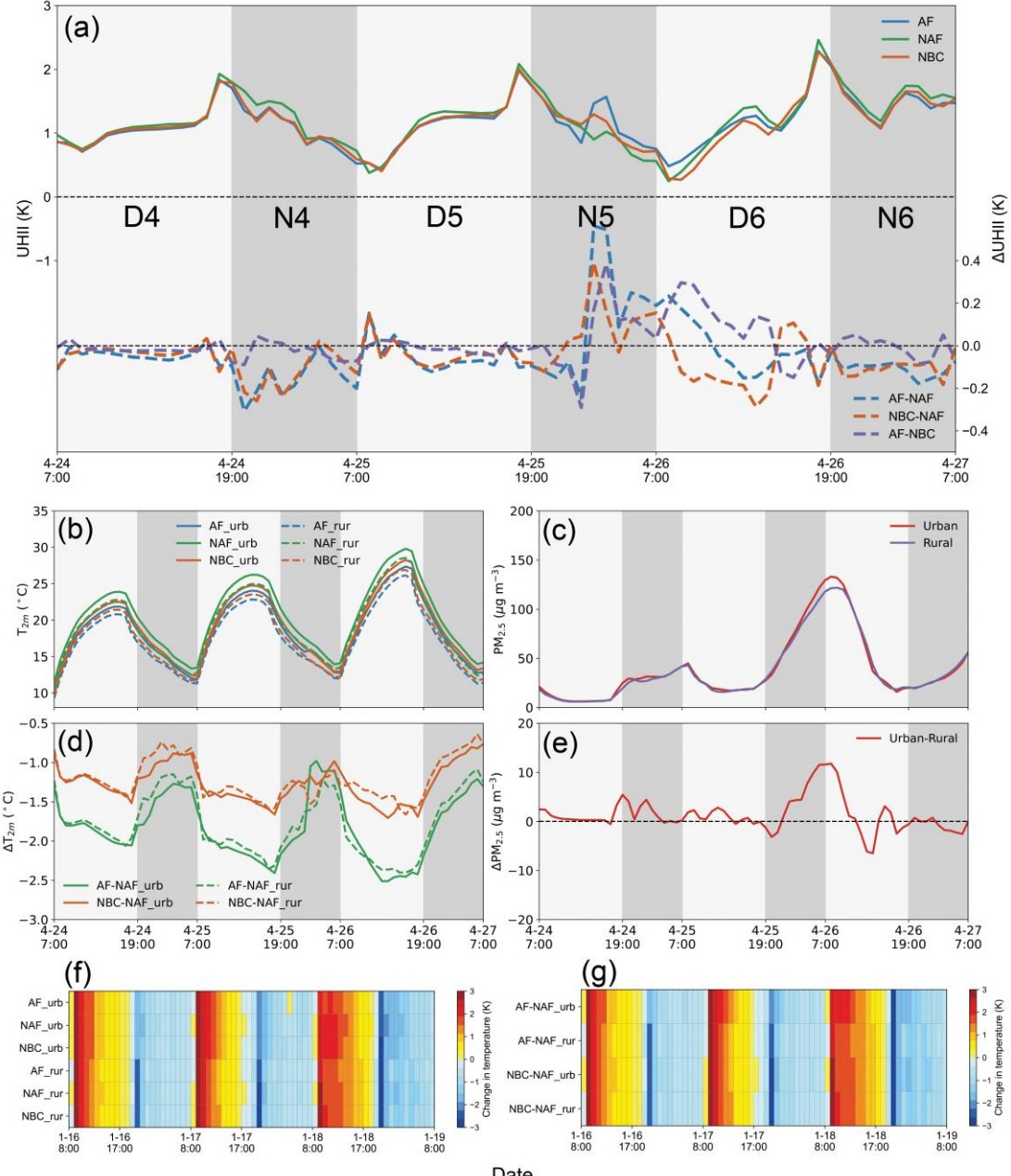

**Figure 5: Same as Fig. 4 but for Case_2018.**

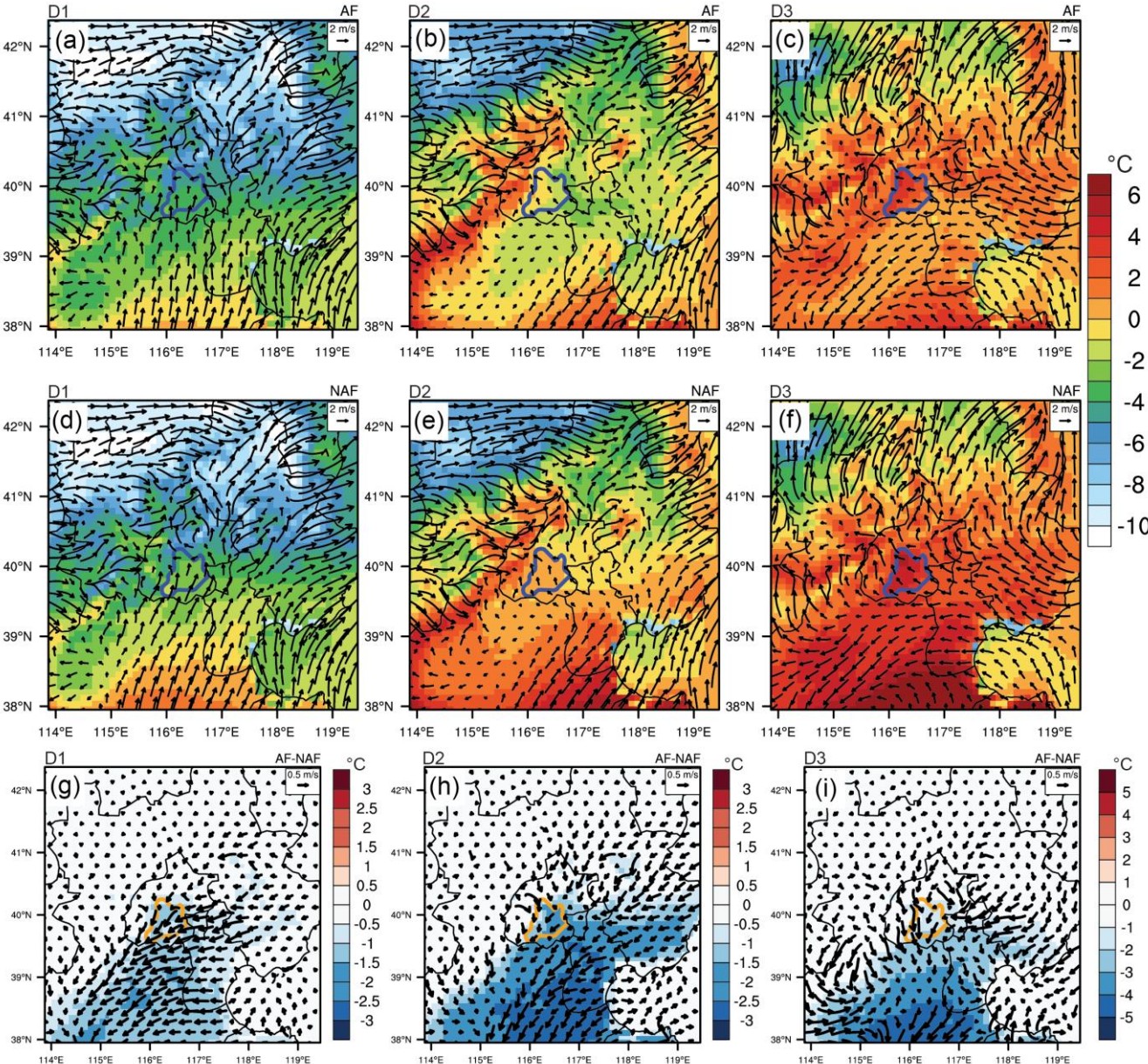

Figure 6: Simulated 2m air temperature and 10m wind field in AF (first row), NAF (second row) and differences between AF and NAF (third row) on D1 (first column), D2 (second column), and D3 (third column). The areas within the blue (a-f) and orange (g-i) line are urban areas of Beijing.