# Peer review of "Circulation-regulated impacts of aerosol pollution on urban heat island in Beijing"

_EGUsphere, 2022_

## Author Comment (AC1)

**Response to Reviewer #1**

This paper presents the results of the aerosol impact on urban heat island (UHI) over Beijing. The authors first analyzed the 2016-2020 observations to link UHI intensity (UHII) with wind direction and $PM_{2.5}$ pollution, then used WRF-Chem regional model to do the perturbation simulations of a haze episode in January 2010 to substantiate the underlying mechanism of such linkage. The general conclusion was that aerosols, either locally emitted or transported and cumulated through regional circulation, reduced UHII via aerosol-radiation interactions over the study region. Though the research topic fits into the ACP scope and the paper was concisely written, more analysis and discussions are needed to reconcile the mismatch of time period and time scale between the observational and modeling analysis to make the conclusion robust under different seasons and aerosol pollution conditions. This is especially important to the regions like Beijing, who has experienced the rapid changes in landscape and pollutant emissions over the past decade. In addition, quite a large portion of the discussion were descriptive and qualitative, while more quantitative analysis should and can be done with the available observation and simulation data.

➢ Reply: Thanks for the careful review and the comments are valuable to enhance the quality of our manuscript.

➢ We have conducted additional simulations and added more discussions to reconcile the mismatch of time period and time scale between the observational and modeling analysis in Section 3.1. The additional simulations are for a light pollution event occurred in the Spring of 2018 to make the conclusion robust under different seasons and aerosol pollution conditions.

➢ We found that due to lower $PM_{2.5}$ concentration, ARE reduces UHII by less than 0.2 K. The lower concentration also diminishes absorption of shortwave radiation during the daytime which reduces downward longwave radiation, leading to weakened UHII in nighttime. We have also added more quantitative analysis in Section 3 (marked in the revised manuscript).

On top of the above comments, the authors are also expected to address the following specific points:

Section 2.1: Can the authors elaborate what is the criteria they chose the weather stations for UHI estimate? This information is important since the selection of urban vs. rural stations may slew the results. Only 2 urban weather stations were selected for the analysis. How representative were they for Beijing?

➢ Reply: We chose these weather stations because only data at these stations in Beijing are available.

➢ We have already clarified in Section 3.1 that our observation-based results are representative for UHI in western and northern sides of Beijing as we used stations there as rural.

➢ "We used rural stations located in the west and north of Beijing as rural in the calculation of UHII, and PM2.5 concentrations are usually much lower there (Fig. S2)."

➢ As a result, we found different results compared with previous research and we further conducted model simulations to understand the underlying mechanism.

Section 3.1 – Fig. 1 discussion: PM$_{2.5}$ data from all observation sites, urban or rural, were selected for daily average calculation to distinguish between polluted vs. clean days. Was there large PM$_{2.5}$ gradient between the urban and rural sites? What was the impact of such PM$_{2.5}$ gradient if it existed?

➢ Reply: PM$_{2.5}$ concentrations in urban stations were higher than those in rural stations by 6.9 μg m$^{-3}$ on average over 2016-2020. Under polluted conditions, the difference reached 14.4 μg m$^{-3}$ on average. Such gradient may result in the overestimation of pollution for those rural stations and change the statistical results.

➢ We further evaluated the results based on the standard that PM$_{2.5}$ concentrations at all stations meet the criterion of clean or polluted (Fig. R1) and the standard that average PM$_{2.5}$ concentration of all urban stations and rural stations should meet the criterion of clean or polluted (Fig. R2).

➢ Compared with Fig. R3 (Fig. 1 in the revised manuscript), we found similar distributions and some minor difference in mean values. When PM$_{2.5}$ concentrations at all stations meet the criterion of clean or polluted, we found the mean values increased by 0.03-0.04 K for clean conditions but decreased by 0.14 K during daytime and 0.06 K during nighttime. When we used average PM$_{2.5}$ concentration of all urban stations and rural stations to determine clean or polluted, mean values decreased by 0.01 K for clean conditions and increased by 0.01 K and 0.06 K during daytime and nighttime, respectively.

➢ The changes are not notable and we added these comparisons in the revised manuscript.

➢ In the revised manuscript, we further added Table R1 to show the distribution of daily average urban and rural PM$_{2.5}$ concentration under clean and polluted conditions.

➢ We found that there were 17.07% overestimation in rural stations because of the gradient between urban and rural areas. However, we also observed that PM$_{2.5}$ concentrations were over 60 μg m$^{-3}$ for most of considered days. Besides, we found pollution in urban and rural shows a good occurrence, and we thus believe using the daily mean PM$_{2.5}$ concentration averaged over all stations can properly represent the regional feature of aerosol pollution.

➢ Therefore, to better characterize the regional air quality and avoid effects of individual station, all stations within the administrative divisions of Beijing were selected to calculate mean PM$_{2.5}$ concentration to distinguish between polluted and clean days.

[Figure]

**Figure R1:** Probability distribution of UHII$_{obs}$ (a, d), UHII$_{max}$ (b, e) and UHII$_{min}$ (c, f) under different pollution conditions. Clean means PM$_{2.5}$ concentrations of all stations are below 75 µg m$^{-3}$. Polluted means PM$_{2.5}$ concentrations of all stations are equal or over 75 µg m$^{-3}$. The bold curve in each subgraph is normal distribution curve, and µ denotes the average value.

[Figure]

**Figure R2:** Probability distribution of UHII$_{obs}$ (a, d), UHII$_{max}$ (b, e) and UHII$_{min}$ (c, f) under different

pollution conditions. Clean means both average $PM_{2.5}$ concentrations of all urban stations and those of rural stations are below 75 µg m⁻³. Polluted means both average $PM_{2.5}$ concentrations of all urban stations and those of rural stations are equal or over 75 µg m⁻³. The bold curve in each subgraph is normal distribution curve, and µ denotes the average value.

[Figure]

**Figure R3:** Probability distribution of $UHII_{obs}$ (a, d), $UHII_{max}$ (b, e) and $UHII_{min}$ (c, f) under different pollution conditions. Clean means average $PM_{2.5}$ concentration of all stations is below 75 µg m⁻³. Polluted means average $PM_{2.5}$ concentration of all stations is equal or over 75 µg m⁻³. The bold curve in each subgraph is normal distribution curve, and µ denotes the average value.

Table R1: Distribution of daily average urban and rural $PM_{2.5}$ concentration (unit: µg m⁻³) under clean and polluted conditions. Here, $PM_{2.5\_average}$ represents average $PM_{2.5}$ concentrations of all stations; $PM_{2.5\_urban}$ represents average $PM_{2.5}$ concentrations of all urban stations; $PM_{2.5\_rural}$ represents average $PM_{2.5}$ concentrations of all rural stations.

|  | $PM_{2.5\_average} \geq 75$ (369 days) | $PM_{2.5\_average} < 75$ (1373 days) |
|---|---|---|
| $PM_{2.5\_urban} \geq 75$ | 366 (99.19%) | 18 (1.31%) |
| $PM_{2.5\_urban} < 75$ | 3 (0.81%) | 1355 (98.69%) |
| $PM_{2.5\_rural} \geq 75$ | 306 (82.93%) | 12 (0.87%) |
| $PM_{2.5\_rural} < 75$ | 64* (17.07%) | 1361 (99.13%) |

*: These 64 days consist of 4 days with $PM_{2.5\_rural} < 50$, 8 days with $50 \leq PM_{2.5\_rural} < 60$ and 52 days with $60 \leq PM_{2.5\_rural} < 75$.

Section 3.1 – Fig. 2 discussion: A figure or table shows the average $PM_{2.5}$ conc. over urban/rural areas under each prevalent wind directions should be provided to support the argument.

➢ Reply: We have added a table shows the average $PM_{2.5}$ concentration over urban and rural areas under each prevalent wind directions in the manuscript.

Table R2. Average $PM_{2.5}$ concentration (unit: µg m⁻³) in urban and rural areas under each prevalent wind

directions.

| Wind directions | Easterly | Southerly | Westerly | Northerly |
|---|---|---|---|---|
| Urban PM$_{2.5}$ | 58.23 | 53.88 | 52.24 | 49.49 |
| Rural PM$_{2.5}$ | 50.82 | 47.34 | 44.68 | 43.31 |

Section 3.2 – in Fig. S2, how did the authors derive the observed time series of UHII? Was it the average UHII at the same days/hours from 2016 to 2020, or other? Since the modeled UHII reflected the heavy polluted condition, why not compared the modeled UHII with the observed one under the pollution condition?

➢ Reply: We conducted model simulations of a typical haze event that occurred in January 2010 in Beijing, and Fig. S2 (Fig. S5 in the revised manuscript) is the comparison done with observations during this haze event.

➢ It is under the heavy polluted condition.

➢ In the revised manuscript, we also added an evaluation of UHII during a light polluted case in 2018 (Fig. R4, Fig. S6 in the revised manuscript).

[Figure]

**Figure R4:** Observed and simulated UHII by AF in Case_2018 in Beijing. Observations are obtained from the stations listed in Table S1.

In Line 158: "…and differences in values are generally within the trusted range". What is the trusted range of UHII comparison? How did modeled wind and temperature compare to the respective observations?

➢ Reply: There is no specific definition for the trusted range of UHII comparison.

➢ However, according to previous explorations, if the model can successfully reproduce the temporal variation of UHII and the mean bias and root mean square error are smaller than 2 K, the results are acceptable.

➢ In our simulation, we successfully reproduce the temporal variation of UHII, and the mean bias and root mean square error are 1.16 K and 1.47 K in Case_2010 and 1.02 K and 1.27 K in Case_2018, respectively.

➢ The modeled wind and temperature for the 2010 case have been shown in our previous work (Gao et al., 2016).

➢ We also added the performance for Case_2018 in Fig. R5 (Fig. S4 in the revised manuscript).

[Figure]

**Figure R5:** Simulated and observed 2m air temperature (a), 10m wind speed (b) and near-ground PM$_{2.5}$ concentration (c) in Case_2018. Observations are obtained from the stations listed in Table S1 and Table S2.

Line 168: how was heat storage calculated?

➤ Reply: In WRF-Chem model, heat storage is calculated with land surface model, and we applied Noah land surface scheme for non-urban grids and Urban Canopy model for urban grids. In Noah land surface scheme, heat storage is calculated using the following equations:

$$G = (1 - F_{veg})G_b + F_{veg}G_v$$

$$G_b = \frac{2\lambda_{isno+1}}{\Delta z_{isno+1}}(T_{g,b} - T_{isno+1})$$

$$G_v = \frac{2\lambda_{isno+1}}{\Delta z_{isno+1}}(T_{g,v} - T_{isno+1})$$

➤ where, $F_{veg}$ denotes fractional vegetated area, $G_b$ and $G_v$ are heat storage for bare ground and vegetated ground, respectively, and $\lambda_{isno+1}$ represents thermal conductivity of the surface layer of snow or soil; $z_{isno+1}$ is layer thickness of the surface layer of snow or soil, $T_{isno+1}$ represents temperature of the surface layer of snow (when $isno+1<0$) or soil (when $isno = 0$), and $T_{g,b}$ and $T_{g,v}$ stand for ground surface temperature at bare ground fraction and vegetated fraction, respectively.

➤ In Urban Canopy model, heat storage is calculated using

$$G = G_0 + 2\int_0^{z_r}\left[\frac{\partial(\rho_b c_b T_b)}{\partial t}\right]d_z$$

➤ where, G$_0$ is the surface heat flux into the ground per unit area, including roof and road, and $\rho_b$, $c_b$, and $T_b$ are density, specific heat, and temperature of buildings.

➤ We have added these descriptions in the revised manuscript.

---

## Author Comment (AC2)

**Response to Reviewer #2**

This manuscript examined the characteristics of UHI in Beijing with observation and model simulation focusing on the variable impacts of aerosol and regional circulation on the intensity of UHI. The authors clearly showed weakened intensity of UHI associated with aerosol pollution both in daytime and nighttime, which is opposed to what has been reported in the previous literatures. They also exhibited the differences in UHI among the different wind directions and tried to reveal the mechanisms behind them with sensitivity simulations using WRF-Chem model. I acknowledge that some of these findings are important not only to the science of UHI but also to co-controlling UHI and urban air pollution issues in the city. This paper is rightfully within the scope of ACP, however, I noticed several issues in this manuscript which cannot be passed over to be published. I suggested that the authors should consider the following comments.

> ➤ Reply: Thanks for the careful reading and valuable comments. We have revised our manuscript accordingly.

Major Comment:

The manuscript was basically well organized, and each chapter (and sub chapter) summarized the information concisely. However, in many aspects. descriptions were too concise to understand properly what they mean. Most of the figure captions were insufficient, and several key points of the manuscript lacked convincing explanations and discussions but rather just cited the previous literatures. These kinds of terrible lack of information largely deteriorate the value of the manuscript. I strongly recommend the author to carefully revise the manuscript, figure and figure captions to make the paper more scientifically readable. I noticed concrete points of revision as below.

> ➤ Reply: Thanks for your kind suggestions, we have revised our manuscripts according to the listed comments.

Specific Comments:

- L117: The definition of UHII is not well described. Did you take the difference between the averaged T2m over all urban stations and all rural stations?

> ➤ Reply: Yes, UHII is the difference between the averaged T2m over all urban stations and all rural stations.
> ➤ We have modified the expression in Section 2.4 to "We defined $UHII_{obs}$, as observed differences in average 2m air temperature ($T_{2m}$) between all urban stations and all rural stations."

- L120-122: You should explain why you choose the different definition of UHII for simulation from that for the observation and how large impact the difference in the definition will have. Since the rural area in calculating the UHII for simulation shown in Fig S4 and S5 is largely different from the area where the rural observation sites are located, I suppose the impact cannot be negligible.

➢ Reply: In Section 3.1 and Sections 3.2, we discussed the impacts concluded through these two definitions.

➢ The observations were used to figure out long-term features of UHII in the northwest direction of Beijing in our study.

➢ To better clarify the influence induced by selection of rural areas, we added Fig. R6 (Fig. S7 in the revised manuscript) to show the simulated UHII calculated based on site locations and area average. Apparent difference can be found that site-based UHII decreases more than area-based UHII especially at nighttime because of lower $PM_{2.5}$ concentrations in the rural sites than selected rural area.

➢ Limited by the spatial distribution of observation sites and observation variables, we used model to explore the underlying mechanism.

➢ We stated at the beginning of Section 3.2: "To understand the underlying mechanism of the varying influences and to reduce uncertainty induced by selection of monitoring stations, we conducted model simulations of a typical haze event that occurred in winter in Beijing."

➢ We also added more explanations in Data and Methods section: "We chose these two different definitions of UHII for observation and simulation to evaluate uncertainty induced by the spatial limitation of monitoring stations.".

[Figure]

**Figure R6:** Variations of site-based and area-based UHII and difference between AF and NAF in Case_2010.

- Fig1: Insufficient caption. What is the bold curve on the figures? What is the definition of μ? Horizontal axis should be UHII (not UHI) for a and d, UHII_max for b and e, UHII_min for c and f.

➢ Reply: The bold curve is the normal distribution curve, μ is the average value. We have changed UHI to $UHII_{obs}$ for a and b, $UHII_{max}$ for b and e, $UHII_{min}$ for c and f.

➢ We have also modified the caption to "Figure 1: Probability distribution of $UHII_{obs}$ (a, d), $UHII_{max}$ (b, e) and $UHII_{min}$ (c, f) under different pollution conditions. The bold curve in each subgraph is the normal distribution curve and μ denotes the average value.".

- L125: Fig.1 is unrelated to wind directions.
  ➢ Reply: Yes, Fig. 1 is unrelated to wind direction.
  ➢ We changed the description to "Fig. 1 presents the probability distributions of UHII under different $PM_{2.5}$ concentrations".

- L126: The definition of clean day and pollution day is not clear. Did you calculate the daily mean $PM_{2.5}$ concentration averaged over all stations regardless of urban and rural and that all-station-averaged value is used for clean/polluted day judgement? Or, every station should pass the criteria to be judged as clean/polluted day?
  ➢ Reply: We used daily mean $PM_{2.5}$ concentrations averaged over all stations.
  ➢ $PM_{2.5}$ concentrations in urban stations were higher than those in rural stations by 6.9 µg m$^{-3}$ on average over 2016-2020. Under polluted conditions, the difference reached 14.4 µg m$^{-3}$ on average. Such gradient may result in the overestimation of pollution for those rural stations and change the statistical results.
  ➢ We further evaluated the results based on the standard that $PM_{2.5}$ concentrations at all stations meet the criterion of clean or polluted (Fig. R1) and the standard that average $PM_{2.5}$ concentration of all urban stations and rural stations should meet the criterion of clean or polluted (Fig. R2).
  ➢ Compared with Fig. R3 (Fig. 1 in the revised manuscript), we found similar distributions and some minor difference in mean values. When $PM_{2.5}$ concentrations at all stations meet the criterion of clean or polluted, we found the mean values increased by 0.03-0.04 K for clean conditions but decreased by 0.14 K during daytime and 0.06 K during nighttime. When we used average $PM_{2.5}$ concentration of all urban stations and rural stations to determine clean or polluted, mean values decreased by 0.01 K for clean conditions and increased by 0.01 K and 0.06 K during daytime and nighttime, respectively.
  ➢ The changes are not notable and we added these comparisons in the revised manuscript.
  ➢ In the revised manuscript, we further added Table R1 to show the distribution of daily average urban and rural $PM_{2.5}$ concentration under clean and polluted conditions.
  ➢ We found that there were 17.07% overestimation in rural stations because of the gradient between urban and rural areas. However, we also observed that $PM_{2.5}$ concentrations were over 60 µg m$^{-3}$ for most of considered days. Besides, we found pollution in urban and rural shows a good occurrence, and we thus believe using the daily mean $PM_{2.5}$ concentration averaged over all stations can properly represent the regional feature of aerosol pollution.

➢ Therefore, to better characterize the regional air quality and avoid effects of individual station, all stations within the administrative divisions of Beijing were selected to calculate mean $PM_{2.5}$ concentration to distinguish between polluted and clean days.

[Figure]

**Figure R1:** Probability distribution of $UHII_{obs}$ (a, d), $UHII_{max}$ (b, e) and $UHII_{min}$ (c, f) under different pollution conditions. Clean means $PM_{2.5}$ concentrations of all stations are below 75 µg m$^{-3}$. Polluted means $PM_{2.5}$ concentrations of all stations are equal or over 75 µg m$^{-3}$. The bold curve in each subgraph is normal distribution curve, and µ denotes the average value.

[Figure]

**Figure R2:** Probability distribution of $UHII_{obs}$ (a, d), $UHII_{max}$ (b, e) and $UHII_{min}$ (c, f) under different pollution conditions. Clean means both average $PM_{2.5}$ concentrations of all urban stations and those of rural stations are below 75 μg m$^{-3}$. Polluted means both average $PM_{2.5}$ concentrations of all urban stations and those of rural stations are equal or over 75 μg m$^{-3}$. The bold curve in each subgraph is normal distribution curve, and μ denotes the average value.

[Figure]

**Figure R3:** Probability distribution of $UHII_{obs}$ (a, d), $UHII_{max}$ (b, e) and $UHII_{min}$ (c, f) under different pollution conditions. Clean means average $PM_{2.5}$ concentration of all stations is below 75 µg m$^{-3}$. Polluted means average $PM_{2.5}$ concentration of all stations is equal or over 75 µg m$^{-3}$. The bold curve in each subgraph is normal distribution curve, and µ denotes the average value.

Table R1: Distribution of daily average urban and rural $PM_{2.5}$ concentration (unit: µg m$^{-3}$) under clean and polluted conditions. Here, $PM_{2.5\_average}$ represents average $PM_{2.5}$ concentrations of all stations; $PM_{2.5\_urban}$ represents average $PM_{2.5}$ concentrations of all urban stations; $PM_{2.5\_rural}$ represents average $PM_{2.5}$ concentrations of all rural stations.

| | $PM_{2.5\_average} \geq 75$ (369 days) | $PM_{2.5\_average} < 75$ (1373 days) |
|---|---|---|
| $PM_{2.5\_urban} \geq 75$ | 366 (99.19%) | 18 (1.31%) |
| $PM_{2.5\_urban} < 75$ | 3 (0.81%) | 1355 (98.69%) |
| $PM_{2.5\_rural} \geq 75$ | 306 (82.93%) | 12 (0.87%) |
| $PM_{2.5\_rural} < 75$ | 64* (17.07%) | 1361 (99.13%) |

*: These 64 days consist of 4 days with $PM_{2.5\_rural} < 50$, 8 days with $50 \leq PM_{2.5\_rural} < 60$ and 52 days with $60 \leq PM_{2.5\_rural} < 75$.

- L127: Unit of UHII should be [K] not [°C]
  ➢ Reply: We have changed all units of UHII to K.

- L130-136: This part is not described well. I cannot fully understand what you want to mean here. According to the description in this part, the strengthened LW radiation in nighttime due to the absorption of sunlight by aerosol in daytime alleviate the temperature reduction in nighttime, which should lead to intensify nighttime UHII in polluted situation compared to clean condition. However, you also state that ARE reduces near surface temperature in urban areas, leading to a weakened UHII ** throughout the day**. Could you explain more about the mechanism how ARE reduce near surface temperature and weaken UHII in nighttime? I guess it's better to consider using schematic diagram to explain the complicated role of aerosol on UHII in daytime and nighttime.
  ➢ Reply: In this part, we intended to explain why our results are different from others. To avoid confusion, we corrected our explanations in the revised manuscript.
  ➢ In this research, we used rural stations located in the west and north of Beijing as rural in the calculation of UHII, and $PM_{2.5}$ concentrations are usually much lower there (Fig. R7, Fig. S2 in the revised manuscript). As a result, temperature at these rural stations is less affected by aerosol pollution.

Daily PM$_{2.5}$ concnetration from 2016 to 2020

[Figure]

**Figure R7.** Distribution of daily mean PM$_{2.5}$ concentration in different concentration intervals over 2016-2020; data at 1002A and 1009A stations are used, and these two stations are located in north of Beijing.

- ➢ We designed a simplified flow chart (Fig. R8, Fig. S3 in the revised manuscript) to show how UHII is changed during daytime and nighttime, assuming rural areas are not influenced by ARE.
- ➢ ARE reduces near surface temperature in urban areas, leading to a weakened UHII and heat storage throughout the day. Although the strengthened longwave radiation process at nighttime due to absorption of aerosols at daytime alleviates the reduction of temperature in urban areas, decreased daytime temperature and heat storage release contribute more to near surface temperature and results in weakened UHII.
- ➢ However, the increase of UHII due to strengthened longwave radiation process is smaller than that by the process during daytime.
- ➢ We have added these explanations in the revised manuscript.

[Figure]

**Figure R8:** Flow chart shows how UHII is changed during daytime and nighttime assuming rural areas are not influenced by ARE. Pink boxes stand for increasing trend while blue boxes show decreasing trend.

- Fig2: Insufficient caption. What is the definition of box-whisker plots?
  - ➢ Reply: Box-whisker plots show distribution of all UHII in different classifications based on wind direction and $PM_{2.5}$ concentration. Box chart values from the bottom to up are the mean value minus one time of standard deviation, 25% quantile line, 75% quantile line, and the mean value plus one time of the standard deviation.
  - ➢ We have modified the caption to "Distribution of UHII under different wind and pollution conditions. Panel (a) and (c) are classified based on the wind direction in urban areas, while panel (b) and (d) are based on wind direction in rural areas. Green triangles represent average values, red lines are median values, the box chart value from the bottom to up is the mean value minus one time of standard deviation, 25% quantile line, 75% quantile line, the mean value plus one time of the standard deviation."

- L140: More words are necessary to explain why the reduction of aerosol in urban area in the case of northerly wind led to elevated UHII. Especially, it's quite confusing that even though the northerly wind reduces the aerosol in urban area, it is still classified as "polluted". So, you should explain why the difference in UHII between clean and polluted conditions is minimal under northerly wind. Just citing Gao et al. (2016) is not enough.
  - ➢ Reply: The northerly wind indeed reduces the aerosol in urban area, but the concentration is relatively reduced compared to southerly region in Beijing. Aerosol concentration in urban areas is still high enough to be classified as polluted.
  - ➢ Fig S11c is an example for such condition, which shows relatively lower $PM_{2.5}$ concentration in northerly regions in Beijing yet the value is still over 100 µg m$^{-3}$.
  - ➢ We modified the expression in the manuscript to "We observe elevated UHII when northerly winds are prevalent in urban areas on polluted days (Fig. 3a, c). The mean UHIIs are 2.0 and 1.8 K at daytime and 2.9 and 2.8 K at nighttime on clean and polluted days, respectively. This is because northerly winds reduce aerosol concentrations in urban areas (Table 1). Although $PM_{2.5}$ concentration in urban areas is relatively reduced, it is still high enough to keep the entire area classified as polluted."

- L140-141: "Decreases" in UHII at daytime, from what?
  - ➢ Reply: Decrease are from clean to polluted conditions. We have specified this in the manuscript:
  - ➢ "Decreases in UHII at daytime can be found from clean to polluted conditions under easterly, southerly and westerly winds conditions"

- L142: More explanation is necessary about how the longwave radiation process can weaken the decrease of UHII in polluted condition compared to

clean condition.

> Reply: The longwave radiation process can weaken the decrease of UHII in polluted condition because absorptive aerosol can release heat at night, which alleviates decreases in surface temperature, especially in urban areas with higher PM$_{2.5}$ concentrations (Cao et al., 2016; Yang et al., 2020).

> We have added this explanation in Section 3.1: "Decreases in UHII at daytime can be found from clean to polluted conditions under easterly, southerly and westerly winds conditions. The decreases are weakened at nighttime due to longwave radiation process, as absorptive aerosols release heat at nighttime to alleviates decreases in surface temperature, especially in urban areas with higher PM$_{2.5}$ concentrations (Cao et al., 2016; Yang et al., 2020).

> Our simulation also confirms this process that ARE-induced enhanced longwave radiation weakens UHII at nighttime (Fig. S3).".

> Cao, C., Lee, X., Liu, S., Schultz, N., Xiao, W., Zhang, M., and Zhao, L.: Urban heat islands in China enhanced by haze pollution, Nat Commun, 7, 12509, https://doi.org/10.1038/ncomms12509, 2016.

> Yang, Y., Zheng, Z., Yim, S. Y. L., Roth, M., Ren, G., Gao, Z., Wang, T., Li, Q., Shi, C., Ning, G., and Li, Y.: PM$_{2.5}$ Pollution Modulates Wintertime Urban Heat Island Intensity in the Beijing‐Tianjin‐Hebei Megalopolis, China, Geophysical Research Letters, 47, https://doi.org/10.1029/2019gl084288, 2020.

[Figure]

**Figure R9**: Variation of simulated UHII and downward longwave radiation at ground surface in Case_2010.

- L145-148: The foehn wind can be a reason for the reversal of thermal gradients under westerly, but it cannot be a reason under southerly, because there are no high mountain ranges in the south of Beijing.

> Reply: The reason for the weak UHII when southerly winds are prevalent

is associated with the warm southerly wind.

➢ We have modified the expression in the manuscript: "The weak UHII under westerly wind condition is associated with foehn wind that northwesterly or westerly travel through the Yan Mountains, as foehn wind is able to heat rural areas and to reduce the urban-rural thermal gradients (Ma et al., 2013). The weak UHII when southerly winds are prevalent is caused by warmer southerly wind from lower latitude, which can only significantly heat southern rural areas because urban impervious surface and buildings have larger heat capacities than rural surface, and tall buildings in the city can block air flow. We also detect larger reductions in UHII when westerly or southerly winds are dominant (Fig. 2b, d), suggesting that foehn wind and warm southerly wind are likely to amplify the weakening effect of pollutants on UHII."

- Fig3 (and FigS5): I was confused, and I could not understand why NBC-NAF is used in these figures. If you want to isolate the BC absorption effect, you should take the difference between NBC and AF. I think NBC-NAF represents the ARE by all aerosols but BC.

➢ Reply: We also discussed the influence of ARE by all aerosols except BC in Section 3.3, so we used NBC-NAF.

➢ We have added AF-NBC in Fig. 3.

-L165-168: Since the lines in Fig3 are too thin and unclear to distinguish each other, I could not recognize well what you write in this part.

➢ Reply: We have modified Fig. 3 to make it clear.

[Figure]

**Figure R10:** Variations of UHIIsim of all cases and difference across them (a) in Case_2010. Variations of T2m (b) and △T2m (d) in urban and rural areas. Variations of PM$_{2.5}$ (c) and △ PM$_{2.5}$ (e) in urban and rural areas of AF case. AF-NAF represents the influence of ARE on UHII. NBC-NAF represents the influence of ARE on UHII by all aerosols but BC. AF-NBC represents the influence of BC absorption on UHII.

- L168-169: I could not recognize what you describe here about FigS3c,d: what is "slower pace"? You should describe more clearly here.
  - ➢ Reply: In Fig. S3c, d, heat storage in rural areas (green lines) is smaller but reaches zero earlier than it in urban areas (red lines), which means "Heat storage of rural areas is smaller but released at a slower pace".
  - ➢ We have modified the expression to "Heat storage of rural areas is smaller but heat is released more slowly, as suggested in Fig. S3 that it is smaller in daytime but reaches zero earlier than it in urban areas".

- L170: What does the "difference" mean here?
  - ➢ Reply: The "difference" here means the difference in heat storage and its releasing process between urban and rural areas. It is associated with the previous sentence "Heat storage of rural areas is smaller but heat is released more slowly, as suggested in Fig. S3 that it is smaller in daytime but reaches zero earlier than it in urban areas, leading to a faster declining of T2m in rural areas than urban areas.".
  - ➢ We have changed the expression to make it clear.

- L182: Why FigS4e here? It is for N2 not for D3.
  - ➢ Reply: It should be subgraph "c" here. We have corrected it.

- L184: Could you explain more precisely how the thermal difference of the atmosphere after sunset can modify downward longwave radiation in nighttime?
  - ➢ Reply: It should     be "The impacts of aerosols on UHII are mainly generated by modified downward longwave radiation (LWD) at nighttime, which influences the thermal difference of the atmosphere maintained after sunset."
  - ➢ We have changed it.

- Fig4: What are the blue contours in the figure? No descriptions can be found in figure caption.
  - ➢ Reply: The blue contours represent urban grids.
  - ➢ We have added descriptions in the caption.

- L190: I cannot understand that the weakened warm southerly wind can reduce the UHII in Beijing, since that kind of change in regional scale circulation will evenly influence both urban and rural area which cannot alter the intensity of UHI (UHI is based on the "difference" between urban and rural area). You should explain more precisely about how the mechanism that regional scale circulation change alter the UHI.
  - ➢ Reply: The warm southerly cannot heat urban and rural areas evenly because of the different surface properties and building.
  - ➢ Urban impervious surface and building have larger heat capacities than rural surface, and tall buildings in the city can block air flow. Under a weakened warm southerly wind condition, this unevenness can be amplified which means only southern rural areas in Beijing can be apparently heated, reducing the UHII in Beijing.
  - ➢ We have added these explanations in the revised manuscript.

- L202: I don't think the situation in Fig4c,f,i agree with the observed least impact of aerosol on UHII under northerly wind in urban area, because the prevailing wind direction in Beijing urban area in Fig4c,f is not northerly but westerly or southwesterly.
  - ➢ Reply: Yes, we did not express properly here.
  - ➢ We have changed the expression to "This situation is similar with the observations that strongest UHII occur and alleviated reduction of UHII by aerosol pollution when urban areas are under northerly winds (Fig.2a, c), which are also caused by the southward high aerosol concentration area."